# Development and validation of a clinical instrument to predict risk of an adverse drug reactions in hospitalized patients

Sara Iasmin Vieira Cunha Lima[1]☉*, Rand Randall Martins[2]☉, Valdjane Saldanha[1]‡, Vivian Nogueira Silbiger[3], Isabelle Cristina Clemente dos Santos[1], Ivonete Batista de Araújo[2]‡, Antonio Gouveia Oliveira[1,2]☉

1 Graduate Program in Pharmaceutical Sciences, Health Sciences Center, Federal University of Rio Grande do Norte, Natal, Rio Grande do Norte, Brazil, 2 Pharmacy Department, Health Sciences Center, Federal University of Rio Grande do Norte, Natal, Rio Grande do Norte, Brazil, 3 Department of Clinical and Toxicological Analysis, Health Sciences Center, Federal University of Rio Grande do Norte, Natal, Rio Grande do Norte, Brazil

☉ These authors contributed equally to this work.
‡ These authors also contributed equally to this work.
* sivclima@gmail.com

**Data Availability Statement:** All relevant data are within the manuscript and its Supporting information files.

## Abstract

### Objective

Development and internal validation of a clinical tool for assessment of the risk of adverse drug reactions (ADR) in hospitalized patients.

### Methodology

Nested case-control study in an open cohort of all patients admitted to a general hospital. Cases of ADR were matched to two controls. Eighty four patient variables collected at the time of the ADR were analyzed by conditional logistic regression. Multivariate logistic regression with clustering of cases in a random sample of 2/3 of the cases and respective controls, with baseline odds-ratio corrected with an estimate of ADR incidence, was used to obtain regression coefficients for each risk factor and to develop a risk score. The clinical tool was validated in the remaining 1/3 observations. The study was approved by the institution's research ethics committee.

### Results

In the 8060 hospitalized patients, ADR occurred in 343 (5.31%), who were matched to 686 controls. Fourteen variables were identified as independent risk factors of ADR: female, past history of ADR, heart rate $\geq$72 bpm, systolic blood pressure$\geq$148 mmHg, diastolic blood pressure <79 mmHg, diabetes mellitus, serum urea $\geq$ 67 mg/dL, serum sodium $\geq$141 mmol/L, serum potassium $\geq$4.9 mmol/L, main diagnosis of neoplasia, prescription of $\geq$3 ATC class B drugs, prescription of ATC class R drugs, prescription of intravenous drugs and $\geq$ 6 oral drugs. In the validation sample, the ADR risk tool based on those variables showed sensitivity 61%, specificity 73% and area under the ROC curve 0.73.

**Funding:** This study was financed in part by the Coordenação de Aperfeiçoamento de Pessoal de Nível Superior - Brasil (CAPES) – Finance Code 001. The funders had no role in study design, data collection and analysis, decision to publish, or preparation of the manuscript.

**Competing interests:** The authors have declared that no competing interests exist.

## Conclusion

We report a clinical tool for ADR risk stratification in patients hospitalized in general wards based on 14 variables.

## Introduction

Adverse drug reactions (ADRs) in hospitalized patients have major impact in morbidity, mortality, health outcomes, economic cost and length of stay [1, 2]. Despite their relevance, there is an inherent difficulty in the nature of ADR that affect their identification. In many cases, ADRs may be confused with symptoms or signs related to the clinical condition of the patient [3, 4]. This imposes a further difficulty in the process of care, as it prevents or delays the appropriate intervention.

ADR also have a significant incidence among hospitalized patients. Estimates for the incidence for ADR in the literature vary depending on the definition used, the setting, the study population and the method for detection. A meta-analysis of 22 prospective studies, published in 2012, estimated as 16.9% the cumulative incidence of ADR during hospitalization [5]. An important step in reducing the incidence of ADR would be to identify those patients who are at increased risk of developing ADR from individual patient variables [6]. Some risk factors have already been described [6–9], such as age, sex, polypharmacy, some medication classes, and kidney disease. A few risk predictions models have also been developed and have been validated, internally or externally, but all were intended for populations with specific characteristics, such as the elderly [8–10] or renal patients [6], limiting the applicability of the instrument in inpatients in general.

The existence of a practical clinical instrument for risk stratification that could identify patients hospitalized in wards of general hospitals at risk for ADR, regardless of age or specific clinical condition, would be of great value to increase patient safety and to guide the care of the medical and the clinical pharmacy team. Thus, the objective of this study was the identification of risk factors for ADR in patients hospitalized in general hospital and the construction and internal validation of a clinical tool predictive of ADR.

## Methods

### Study design and population

This was an observational, analytical, case-control study nested in an open cohort consisting of all patients admitted between June 2016 and December 2017 at the Onofre Lopes University Hospital in Natal, Brazil, a tertiary care medium-size university hospital with 247 beds. Male and female patients over 18 years-old, with a length of stay of more than 24 hours and administered at least one medication during the hospitalization were included in the study. Only patients hospitalized in the following departments were included (the hospital is organized in clinical specialty wards and there is no general medical ward): neurology, mental health, nephrology, urology, cardiology, oncology (not receiving chemotherapy), gastroenterology, rheumatology and surgery. Patients admitted because of an ADR, hospitalized in the intensive care unit (ICU), transplanted, under chemotherapy, and pregnant women were excluded. Only the first hospitalization since study start was considered, duplicate hospitalization episodes of the same patients being easily detected because each patient receives a lifetime numeric code that identifies the patient across multiple hospital admissions. The study was

approved by the Research Ethics Committee of the Federal University of Rio Grande do Norte (CAAE 34282914.0.0000.5992) and all patients gave informed consent in written.

## Data collection

Adverse drug reaction was defined, according to the World Health Organization, as a response to a medicine which is noxious and unintended, and which occurs at doses normally used in man for the prophylaxis, diagnosis, or therapy of disease, or for the modification of a physiological function. According to this definition, non-adherence to the treatment, overdose either accidental or intentional, treatment failure and administration errors are not considered ADR [11].

The identification of suspected ADR was carried out on a daily basis through active search in all patients admitted to the wards eligible for this study. The search was conducted by three clinical pharmacists (S.I.V.C.L, V.S, I.B.A.), helped by four previously trained pharmacy students. Before the study start, a list of ADR indicators was created. This was a list of prescription changes that could indicate medical actions in response to an ADR, which was based on the recommendations of the Medication Module Triggers of the IHI Global Triggers Tool [12]. The active search for ADR was performed by inspection of all medication orders for the presence of ADR Indicators, by analysis of the clinical evolution of the patient in the patient charts, and by reviewing nursing charts, as well as through inspection of changes in the prescription and abnormal findings in laboratory tests. The adjudication of ADRs was done independently by two clinical pharmacists of the research team (S.I.V.C.L., I.B.A.), and in case of disagreement, a third researcher (V.S) was consulted.

Each suspected ADR was then assessed for causality using the Liverpool Adverse Drug Reaction Causality Assessment Tool [13], an instrument directed to clinical studies that classifies ADR for causality as definite, probable, possible and improbable. Only ADR classified as definite, probable or possible were considered in this study. The severity of the ADR was classified according to the modified Hartwig's Severity Assessment Scale [14], who classifies ADR as mild (does not require treatment and does not prolong hospitalization), moderate (requires therapy modification, specific treatment or increase in at least one day in the length of stay), severe (potentially life-threatening, causes permanent damage or requires intensive medical care) and lethal (directly or indirectly contributed to the death of the patient). Only ADR of moderate, severe and lethal severity were considered in this study. One researcher (S.I.V.C.L.) was designed to perform all causality and severity classification, in order to reduce bias and standardize the classification.

All patients with suspected ADR were included in the study as cases at the time of the event. The same day, two controls were randomly selected among all patients hospitalized in the eligible wards hospitalized within 5 days of the admission date of the case. In the nested case-control design, one patient may be selected as a control and later become a case, and the same patient may be selected as a control more than once. The patients only remained in the cohort while they did not experience an ADR. After becoming a case, the patient was withdrawn from the study and, therefore, only the first ADR occurrence was considered and no further hospitalizations were considered. Because of the medium size of the hospital, it was felt that it would not be practical to select more than two controls for each case. The nested case-control design was selected because it allows estimation of the risk of ADR from the patient's data at any time throughout the patient's hospitalization period, while traditional risk factor designs only consider risk factors observable at an index moments, usually the time of hospital admission.

The following variables were collected from all cases and controls at the moment the ADR was detected: age, sex, race, self-reported daily intake of alcohol converted to grams of ethanol,

self-reported smoking pack-years, user of intravenous substance, number of previous hospitalizations, type of admission (medical, emergent or elective surgery), previous history of ADR, body mass index, International Classification of Diseases version 10 (ICD-10) chapter of the main diagnosis, comorbidities composing the Charlson's Comorbidity Index [15] (myocardial infarction, chronic heart failure, peripheral vascular disease, cardiovascular disease, dementia, COPD, connective tissue disease, peptic ulcer, diabetes mellitus (without and with target organ damage), moderate to severe kidney disease, tumor, metastases, liver disease (mild and moderate), Charlson's Comorbidity Index, vital signs (blood pressure, heart rate, respiratory rate and body temperature), Glasgow Coma Score, assisted ventilation, urine output, laboratory data (hemoglobin, leukocytes, platelets, serum urea, serum creatinine, sodium, potassium, bicarbonate, albumin, AST, ALT, gamma-GT, alkaline phosphatase, total bilirubin, INR), total number of drugs prescribed, number of drugs prescribed for each of the first level of the Anatomical Therapeutic and Chemical (ATC) Classification system [16], number of drugs prescribed per route of administration, number of drug interactions with C, D and X risk [17], and number of pharmaceutical incompatibilities. In all, 84 variables were collected.

## Statistical analysis

A sample size of 340 cases would afford 80% power, at a significance level of 5%, to identify associations, by conditional logistic regression in a matched case-control study with a 2:1 ratio of controls to cases, with an odds-ratio (OR) of ADR of 1.7 in variables occurring in 5% of the controls to an OR of 1.3 in variables occurring in 50% of the controls [18].

In order to avoid the issue of the non-normal distribution of most interval-scaled predictor variables, all such variables were dichotomized by selecting as cutoff the variable value corresponding to the highest value of the Youden's index computed for each value of the predictor variable [19]. Youden's index is equal to sensitivity + specificity– 1.

For the identification of risk factors for the occurrence of ADR, we initially performed univariate analysis of each predictor variable using conditional logistic regression on the full study population, the dependent variable being the occurrence of an ADR during hospital stay. Results are presented as odds-ratios (OR) with 95% confidence intervals (CI).

The dichotomized variables that in univariate analysis showed a significant association with the occurrence of ADR at a two-tailed significance level <0.10 were included into a multivariate conditional regression model. Variables that had a prevalence among the cases less than 5% and those that exhibited collinearity with other variables, defined as a tetrachoric correlation coefficient $\geq$ 0.70, were excluded. Laboratory parameters with less than 15% missing values had missing values imputed by multivariate sequential imputation using chained equations. Regression coefficients were estimated from the multiply imputed data and those variables showing association with the occurrence of ADR at the two-tailed 5% significance level were selected. Results are presented as adjusted OR (aOR) with 95% CI and p-values.

For the development of a tool for risk prediction of ADR, the 1/3 split-sample method was used, whereby the patient sample was randomly divided into a development sample, consisting of two-thirds of the cases and the respective controls, and a validation sample consisting of the remaining observations. For this purpose, using a computer random number generator each case was assigned a number and the lower 2/3 were selected for the development sample, along with the matching controls.

For the development of a risk stratification tool it is necessary to select a set of predictor variables and to create a scoring system. Our aim was to create the scoring system from the regression equation but, although the conditional logistic model is fully adequate for identification of risk factors and for estimation of odds-ratios in matched case-control designs, the model

does not have a regression constant and thus does not enable the calculation of the probability of ADR given the set of predictors. Unconditional logistic regression with clustering on cases and robust standard errors relaxes the requirement of independence of the observations and is an acceptable method for analysis of matched observations. Therefore, for the development of the scoring system, we used only the data of the development sample and a multivariate binary logistic regression model with clustering on cases and robust standard errors, in which the independent variables were the risk factors previously identified in conditional logistic regression, to estimate the partial regression coefficients. However, the logistic regression equation in case-control designs does not allow the computation of the probability of the outcome, because the regression constant no longer estimates the baseline odds in the target population. Nevertheless, it is possible to correct the regression constant by adding $\ln[\pi/(1-\pi)]-\ln[p/(1-p)]$, where $\pi$ is the population proportion of ADR, estimated from the entire patient cohort, and $p$ the sample proportion of ADR. The probability of ADR in each patient is computed as $1/(1+\text{exponential of}-U_i)$, with $U_i$ being the logit of ADR in the $i$th patient predicted by the regression equation $U_i = b_0 \, x_i b$, where $x$ is the set of predictors, each multiplied by the corresponding partial regression coefficient $b$, and $b_0$ is the regression constant corrected as shown above [20].

To obtain the cut-off for classification of patients at risk of ADR, multiple imputation estimates of the probability of ADR were obtained and, after correcting the value of the regression constant as shown above, the cut-off value was set at the predicted probability corresponding to the maximum value of the Youden's index. The logistic regression coefficients were then rounded to integer values in order to obtain a simplified scoring system that could be applied at the bedside.

Model fit in the development sample was evaluated for calibration with the Hosmer-Lemeshow test and the performance of the prediction model was assessed with the Brier score [21]. The scoring system was assessed for discrimination by the area under the ROC curve (AUC), which was tested for a value of 0.5 corresponding to an indifferent classification system. Model sensitivity and specificity were computed.

For assessment of the internal validation of the ADR risk stratification tool thus developed, the sensitivity, specificity, and AUC of the scoring system were obtained in the validation sample. Under the assumption that patients in whom laboratory parameters have not been ordered are likely to have normal values of those parameters, considering that patient data are not collected at patient admission, but anytime during hospital stay, the ADR risk stratification tool was also evaluated for discrimination coding missing values as 0. Statistical analysis was done with Stata 15 (Stata Corp., College Station, USA).

## Results

### Characteristics of the study population

During the 18-month study period, there were 8060 hospitalization episodes from 6465 unique patients. The mean age of the entire patient cohort was 52.3 ± 17.7 years, 58.8% were female, mean length of stay was 10.4 ± 19.5 days, and the in-hospital mortality rate was 5.70%. The cumulative incidence of ADR was 5.31% (95% confidence interval (CI) 4.77 to 8.88%).

A total of 343 occurrences of ADR were identified and included in the study as cases. Data from 686 matched controls in a 1:2 proportion were obtained, corresponding to 305 distinct patients. No patient declined participation in the study. The average difference between cases and controls in number of days from hospital admission to randomization was 1.90±1.74 days. The wards with greater occurrence of ADR were cardiology and surgery. The characteristics of the study population are shown in Table 1. There were no statistically significant differences

**Table 1. Characteristics of the study population.**

| Variable | Controls n = 686 | Cases n = 343 |
|---|---|---|
| Age, years | 54.7 ± 18.0 | 57.6 ± 17.6 |
| Female sex | 345 (50.3%) | 203 (59.2%) |
| Number of previous admissions | 1.72 ± 1.17 | 1.72 ±1.13 |
| Type of admission | | |
| medical | 407 (59.3%) | 217 (63.5%) |
| emergency surgery | 42 (6.12%) | 25 (7.31%) |
| elective surgery | 237 (34.6%) | 100 (29.2%) |
| Time since admission, days | 13.0 ±10.3 | 12.9 ±10.4 |
| In-hospital mortality | 76 (11.1%) | 44 (12.8%) |
| Ward | | |
| Cardiology | 195 (28.4%) | 126 (36.7%) |
| Surgery | 162 (23.6%) | 65 (19.0%) |
| Gastroenterology | 114 (16.6%) | 23 (6.71%) |
| Nephrology | 58 (8.45%) | 33 (9.62%) |
| Neurology | 55 (8.02%) | 18 (5.25%) |
| Pneumology | 33 (4.81%) | 19 (5.54%) |
| Other | 68 (9.91%) | 59 (17.2%) |

Values are mean ± standard deviation or number(percent).

between groups in the time since admission (p = 0.54) and in-hospital death (p = 0.39). However, the distribution by hospital wards was statistically different (p = 0.001).

Hypoglycemia related to insulin administration was the most common ADR (87, 25.4%), followed by hypotension (68, 19.8%) due to the use of different pharmacological classes of antihypertensive, highlighting the angiotensin-converting enzyme inhibitors (26, 7.6%) and loop diuretics (14, 4.1%). The occurrence of nausea and vomiting was observed in 13.1% (45) of the cases, related to several pharmacological classes such as opioid analgesics (18, 5.2%), general anesthetics (4, 1.2%) and laxatives (4, 1.2%). The other ADRs and medications involved can be seen in the S1 Table.

## Risk factors of ADR

By univariate analysis based on the whole study population, 38 variables associated with the occurrence of ADR were identified (Table 2). For multivariate analysis, 12 variables were excluded, because of collinearity (height, creatinine, type C drug-drug interactions, prescription of ATC class A drugs, and number of medications), prevalence among cases less than 5% (serum bicarbonate, prescription of ATC class H drugs, and main diagnosis of ICD-10 chapter V), missing data (albumin, gamma GT and alkaline phosphatase) and Charlson's comorbidity index to avoid having a score within a score.

After stepwise conditional logistic regression, 14 variables remained statistically significantly associated with the occurrence of ADR (Table 2): female (aOR 1.50), previous history of ADR (aOR 2.05), heart rate ≥ 72 bpm (aOR 1.96), systolic blood pressure ≥148 mmHg (aOR 1.70), diastolic blood pressure < 79 mmHg (aOR 1.96), diabetes mellitus (AOR 2.10), serum urea ≥ 67 mg/dL (aOR 1.94), serum sodium ≥ 141 mmol/L (aOR 1.83), serum potassium ≥ 4.9 mmol/L (aOR 1.67), main diagnosis classified in ICD-10 chapter II—Neoplasms (aOR 2.90), prescription of ≥ 3 ATC class B drugs—blood and blood forming organs (aOR 1.82),

**Table 2. Variables associated with the occurrence of ADR in patients hospitalized in a general hospital in univariate and multivariate analysis by conditional logistic regression.**

| Variable | Univariate analysis | | | Multivariate analysis | | |
|---|---|---|---|---|---|---|
| | OR | 95% CI | | p | aOR | 95% CI | | p |
| Age > 58 years | 1.60 | 1.23 | 2.07 | <0.001 | | | | |
| Female | 1.54 | 1.11 | 2,13 | 0.009 | 1.50 | 1.09 | 2.07 | 0.012 |
| Past history of ADR | 2.11 | 1.51 | 2.93 | <0.001 | 2.05 | 1.39 | 3.03 | <0.001 |
| Weight ≥ 73 Kg | 0.67 | 0.50 | 0.90 | 0.008 | | | | |
| Height ≥ 159 cm | 0.75 | 0.57 | 0.99 | 0.048 | - | - | - | - |
| Heart rate ≥ 72 bpm | 1.56 | 1.17 | 2.09 | 0.003 | 1.96 | 1.37 | 2.80 | <0.001 |
| Systolic blood pressure ≥ 148 mmHg | 1.78 | 1.19 | 2.67 | 0.005 | 1.70 | 1.01 | 2.86 | 0.045 |
| Diastolic blood pressure < 79 mmHg | 1.47 | 1.06 | 1.11 | 0.008 | 1.96 | 1.35 | 2.84 | <0.001 |
| Glasgow score ≥14 | 0.48 | 0.25 | 0.93 | 0.030 | | | | |
| Diabetes mellitus | 2.57 | 1.94 | 3.10 | <0.001 | 2.10 | 1.50 | 2.96 | <0.001 |
| Kidney disease | 1.39 | 1.02 | 1.91 | 0.039 | | | | |
| Liver disease | 0.44 | 0.26 | 0.74 | 0.002 | | | | |
| Charlson's index ≥ 4 | 1.66 | 1.27 | 2.17 | <0.001 | - | - | - | - |
| Hemoglobin < 12 g/dL | 1.54 | 1.16 | 2.03 | 0.002 | | | | |
| Urea ≥ 67 mg/dL | 2.35 | 1.75 | 3.18 | <0.001 | 1.94 | 1.33 | 2.82 | 0.001 |
| Creatinine ≥ 1.4 mg/dL | 2.13 | 1.59 | 2.86 | <0.001 | - | - | - | - |
| Sodium ≥ 141 mmol/L | 1.61 | 1.03 | 2.53 | 0,036 | 1.83 | 1.07 | 3.15 | 0.028 |
| Potassium ≥ 4.9 mmol/L | 2.47 | 1.80 | 3.41 | <0,001 | 1.67 | 1.11 | 2.20 | 0.013 |
| Albumin ≤ 3.1 g/dL | 2.50 | 1.69 | 3.70 | <0.001 | - | - | - | - |
| Gamma GT ≥ 36 UI/L | 1.57 | 1.14 | 2.17 | 0.006 | - | - | - | - |
| Alkaline phosphatase ≥ 85 U/L | 2.53 | 1.59 | 4.02 | <0.001 | - | - | - | - |
| Bicarbonate ≥ 27 mmol/L | 5.62 | 1.14 | 27.8 | 0,034 | - | - | - | - |
| Main diagnosis classified in Chapter II ICD-10—Neoplasms | 2.02 | 1.31 | 3.11 | 0.001 | 2.90 | 1.71 | 4.91 | <0.001 |
| Main diagnosis classified in Chapter V ICD-10—Mental and behavioral disorders | 3.33 | 1.21 | 9.17 | 0.020 | - | - | - | - |
| Main diagnosis classified in Chapter IX ICD-10—Diseases of the circulatory system | 1.49 | 1.10 | 2.02 | 0.009 | | | | |
| Main diagnosis classified in Chapter XI ICD-10—Digestive system disease | 0.40 | 0.25 | 0.64 | <0.001 | | | | |
| Drug interactions ≥7 | 2.04 | 1.53 | 2.74 | <0.001 | | | | |
| Risk C drug interactions ≥ 3 | 2.21 | 1.66 | 2.93 | <0.001 | - | - | - | - |
| Risk D drug interactions ≥ 2 | 2.28 | 1.67 | 3.11 | <0.001 | | | | |
| Prescribed drugs ≥ 8 | 2.76 | 2.05 | 3.72 | <0.001 | - | - | - | - |
| ATC A drugs—Alimentary tract and metabolism ≥ 2 | 1.95 | 1.45 | 2.63 | <0.001 | - | - | - | - |
| ATC B drugs—Blood and blood forming organs ≥ 3 | 1.79 | 1.36 | 2.37 | <0.001 | 1.82 | 1.16 | 2.87 | 0.010 |
| ATC C drugs—Cardiovascular system ≥ 3 | 2.18 | 1.64 | 2.89 | <0.001 | | | | |
| Number of ATC H drugs—Systemic hormonal preparations, excluding sex hormones and insulins ≥ 2 | 3.67 | 1.36 | 9.91 | 0,010 | - | - | - | - |
| Prescription of ATC R drugs—Respiratory system | 1.66 | 1.14 | 2.41 | 0,008 | 1.89 | 1.20 | 2.98 | 0.006 |
| Prescription of intravenous drugs | 1.56 | 1.17 | 2.09 | 0,003 | 1.44 | 1.02 | 2.03 | 0.039 |
| Prescription of subcutaneous drugs ≥ 2 | 2.43 | 1.74 | 3.37 | <0.001 | | | | |
| Prescription of oral drugs ≥ 6 | 2.02 | 1.53 | 2.68 | <0.001 | 1.52 | 1.06 | 2.17 | 0.021 |

OR: odds-ratio; aOR: adjusted odds-ratio; -: not included in multivariate analysis.

All variables were collected at the time of the ADR in the cases and of randomization in the controls.

prescription of ATC class R drugs—respiratory system (aOR 1.89), prescription of intravenous drugs (aOR 1.44), and prescription of ≥ 6 oral drugs (aOR 1.52). There was no evidence of collinearity among those variables, the median value of the tetrachoric coefficient, a measure of correlation between binary variables, was 0.10 and the maximum value was 0.56.

## ADR risk stratification tool

For the development of a scoring system for the ADR risk stratification tool, those 14 variables were included into a logistic regression model with clustering of cases and robust standard errors, which was applied to the development sample. After defining a cut-off for prediction of the occurrence of ADR, based on the highest Youden's index, and rounding the regression coefficients, each predictor variable was assigned a number of points derived from the value of its partial regression coefficient. Table 3 displays the partial regression coefficients of each predictor variable, the value of the regression constant after correction, and the points assigned to each item in the ADR risk tool. A score of 29 or more obtained by summation of the points predicts the occurrence of an ADR. The total score of the tool ranges between 0 and 87. In the development sample, the mean score was 26.4 ± 10.4 (limits 0 to 60).

The Hosmer-Lemeshow test was non-significant (p = 0.99) indicating adequate calibration of the model. The area under the ROC curve (Fig 1) was 0.76 (95% CI 0.72–0.79, p<0.001). The sensitivity of the ADR risk tool using the ≥ 22 points threshold was 64.2% and the specificity was 76.1%. The Brier score, which is the mean squared difference of the predicted probability of the outcome to the actual outcome, had a low value of 0.18, indicating adequate predictive performance (a model with 100% correct predictions has a score of 0, with 100% incorrect predictions a score of 1, and a model classifying indifferently a score of 0.25).

When applied to the validation sample, these statistics did not change appreciably: the sensitivity was 64.5%, the specificity 72.6% and the area under the ROC curve 0.74 (Fig 1).

In order to allow the use of the ADR risk stratification tool when laboratory values are not available, urea, sodium and potassium are given 0 points if laboratory results are unavailable. With this new score, the discrimination was maintained (Table 4 and Fig 2).

## Discussion

In this nested case-control study based on a large cohort of patients hospitalized in a general tertiary care hospital we identified 14 independent risk factors of ADR from 84 candidate

**Table 3. Independent risk factors of DRP in patients hospitalized in a general tertiary care hospital and corresponding points in the ADR risk stratification tool.**

| Risk factor | Regression coefficient | Points |
|---|---|---|
| Female | 0.5208062 | 5 |
| Prior history of ADR | 1.119142 | 11 |
| Main diagnosis of ICD-10 chapter II: Neoplasms | 1.000545 | 10 |
| Diabetes mellitus | 0.6876321 | 7 |
| Heart rate ≥72 | 0.6963432 | 7 |
| Systolic blood pressure ≥148 mmHg | 0.978518 | 10 |
| Diastolic blood pressure <79 mmHg | 0.5051598 | 5 |
| Urea ≥67 mg/dL | 0.3848191 | 4 |
| Sodium ≥141 mmol/L | 0.7205159 | 7 |
| Potassium ≥4.9 mmol/L | 0.5206034 | 5 |
| Prescription of ATC class B drugs (Blood and blood forming organs) ≥3 | 0.400401 | 4 |
| Prescription of an ATC class R drug (Respiratory system) | 0.3143512 | 3 |
| Prescription of an intravenous drug | 0.3650177 | 4 |
| Prescription of oral drugs ≥6 | 0.5053098 | 5 |
| Regression constant (corrected) | 2.9394309 | |

A total score ≥ 29 points predicts the occurrence of an ADR.

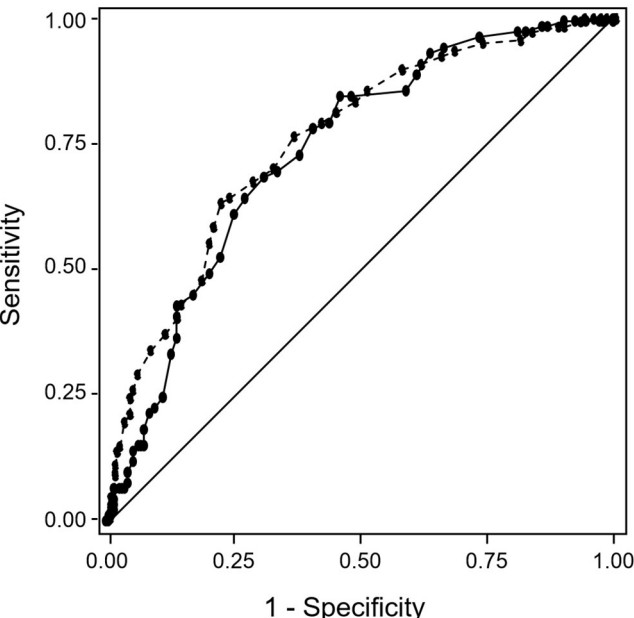

**Fig 1. ROC curves of the ADR risk stratification tool in the development sample (dashed line) and in the validation sample (solid line).**

variables describing patient characteristics, clinical diagnoses, and medication details (number administered, ATC drug class, administration route, drug incompatibilies, drug-drug interactions). These risk factors were used to develop a clinical tool for the identification of patients at high risk of ADR that demonstrated good calibration and discrimination ability, especially considering the challenging task of predicting events occurring at a very low rate. The risk tool score can be easily computed at bedside at any time during the hospital stay, to help identify those patients in a general medical or surgical ward who are at risk of experiencing an ADR.

It is observed that almost all RAM are typical of the main drugs used, such as hypoglycemia and hypotension associated with hypoglycemic and antihypertensive drugs, respectively. Insulin is commonly related to adverse reactions like this in the hospital setting [22]. The drug-dependent dose reactions may occur due to limitations, such as the controlled diet with limited carbohydrates in the hospital environment and bed restriction [23], resulting in the particularity of the individual dose, subject to the need for constant dose adjustments [24, 25]. Hypotension also seems to be related to the patient's fragility, length of hospital stay and, mainly, age and concomitant use of more than one antihypertensive [26, 27].

To the best of our knowledge, this is the first clinical tool developed for risk stratification of ADR among patients hospitalized in general hospitals. We have found four models published the last 10 years on risk prediction of ADR. In all of them the predictive models were

**Table 4. Model performance statistics in the development and the validation samples of the ADR risk stratification tool with unavailable laboratory values scored 0.**

| Statistic | Development sample | | | Validation sample | | |
|---|---|---|---|---|---|---|
| | n | % | 95% CI | n | % | 95% CI |
| Sensitivity | 141/229 | 61.6 | 54.9–67.9 | 70/114 | 61.4 | 51.8–70.4 |
| Specificity | 362/458 | 79.0 | 75.0–82.7 | 171/228 | 75.0 | 68.9–80.5 |
| AUC | 687 | 76.2 | 72.8–79.3 | 342 | 73.2 | 68.1–77.7 |

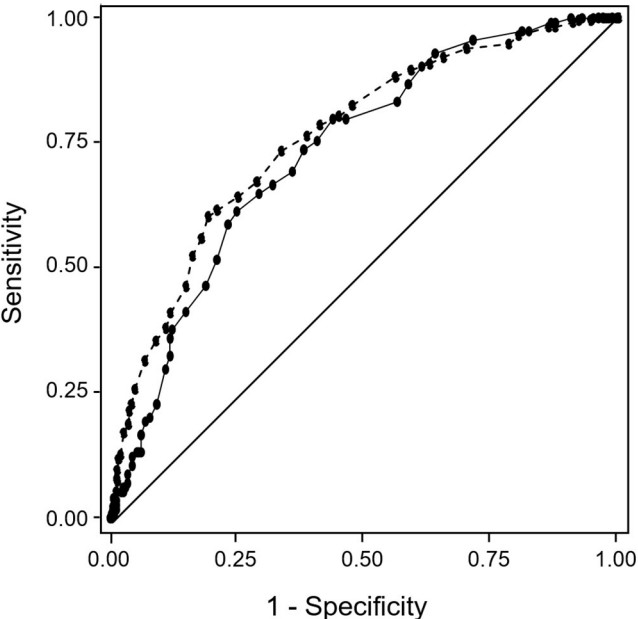

**Fig 2. ROC curves of the ADR risk stratification tool with unavailable laboratory values scored 0 in the development sample (dashed line) and in the validation sample (solid line).**

developed for specific populations, namely elderly patients [8, 10], patients with chronic kidney disease [6], and surgical patients [28]. Two risk prediction tools, the Gerontonet Risk Score [8] and the BADRI Model [10], were validated in an external cohort [9]. The other two presented results only of the internal validation based on bootstrap estimates on the entire patient sample.

Some of the independent risk factors of ADR found in this study have also been described in the literature. Female sex, for example, has been described as a risk factor in a recent systematic literature review [29] and other reports [6, 24]. A previous history of ADR has also been reported as an independent risk factor [8]. Poor renal function, which translates in our findings to increased serum urea, has been reported as a risk factor in several studies [8, 29, 30]. Vascular disease was described as a risk factor for ADR [6] but low diastolic blood pressure has not previously been identified as an independent risk factor, although risk factors related to hypotension have been reported, namely administration of anti-hypertensive and diuretics [31].

Polypharmacy has been often reported as a risk factor of ADR [6, 8, 10], and this result was observed in our study as well. In addition to the administration of 6 or more oral medicines, our results identified two ATC classes of drugs associated with increased risk of ADR: ATC class B drugs, mainly involving anti-plaquetary and anti-coagulant medicines, and ATC class R, respiratory system drugs, which have not been reported yet, although one study identified chronic obstructive pulmonary disease as a risk factor [32].

Besides increased serum urea levels, we identified other laboratory risk factors. Increased serum potassium has previously been reported in the literature [32], but increased sodium levels, such as is found in dehydration, has not been reported previously.

One important risk factor in our study was a diagnosis of diabetes mellitus, most likely because of the predisposition to hypoglycemia, which has not been reported previously, although one study did report administration of anti-diabetic drugs [10]. A diagnosis of

neoplasia has not been reported previously. It must be noted that patients under chemotherapy were not included in our study, therefore the increased risk of ADR in patients with neoplasia is related to the administration of other drugs with increased potential for ADR.

Three other risk factors that were found in this study have not been reported previously: increased heart rate, increased systolic blood pressure, and administration of intravenous drugs. The first is probably related to cardiovascular disease, and several studies have reported cardiovascular risk factors, including chronic heart failure [8], administration of cardiovascular drugs [31, 32], and angina [32]. Increased systolic blood pressure is possible indicating advanced age, which has been identified as risk factor in other studies [6, 29]. Intravenous administration of drugs, because of eventual difficulty in dose adjustment, drug incompatibilities and interactions, and the toxicity profile of many drugs for i.v. administration, is not surprising as risk factor of ADR.

The usual approach to the development of predictive clinical instruments of ADR has been to collect a set of patient variables at the time of hospital admission, which will be used to develop a score that identifies patients who, throughout their hospital stay, are at increased risk of ADR. These models do not appear to be realistic, since a patient's therapy and clinical condition can vary greatly during the hospital stay. Therefore, our option was for a case-control design nested in an open cohort consisting of all patients hospitalized during the study period. Hence, the collected data are those observed at the time of ADR and the clinical tool that was developed in this study stratifies patients by the risk of suffering an ADR in any day of their hospital stay. Nested case-control studies are typically based on closed cohorts, with all subjects admitted to the cohort at the same moment in time, while ours was an open cohort with patients entering the cohort throughout the duration of the study. In order to account for this difference, the matched controls were selected at random from all patients in the eligible wards with approximately the same time as the index case since hospital admission.

Two other studies [30, 31] also adopted the nested case-control design for identification of risk factors of ADR, but have not gone beyond the identification of risk factors and did not attempt to develop an ADR risk prediction tool. The reason was that conditional logistic regression, the method of election for the analysis of matched case-control studies, does not directly provide predicted probabilities of the dependent variable. We went further and used logistic regression adjusted for clustering of cases to obtain the regression coefficients and approximate predicted probabilities of ADR that enabled the development of a scoring system for a risk stratification tool. We first obtained evidence, from the multivariate conditional logistic regression model, of the set of variables that are important predictors of ADR. For the identification of the risk factors, we used the full patient sample as has been recommended by several authors, who have criticized the split-sample method for risk factor identification as a waste of data [33, 34]. In the model specification, we omitted collinear variables in the multivariate model by excluding variables with important correlation with other variables, avoided variables with low prevalence among the cases, and using scores within scores. Throughout the data collection period of the study, we kept record of all patients admitted to the hospital wards included in the study because, in order to be able to use logistic regression in a nested case-control study, it is necessary to estimate the cumulative incidence of the event in the population to correct the value of the regression constant.

Our method of assigning points to the predictor variables and selecting the cut-off point for the total score was different from the methods adopted by some authors, who either assign one point to each risk factor, or round the odds-ratio associated with each predictor and sum them all. We first identified the cut-off point using the Youden's index, a commonly used criterion to define the cut-off point of a continuous variable. Thus, a patient is predicted to have an ADR if the regression equation is equal to or greater than the cut-off. The equation is solved

for that value of the cut-off and, to obtain a simplified score, all partial regression coefficients are rounded to integers.

The internal validation of this tool, based on the 1/3 split sample method, demonstrated very satisfactory discrimination and predictive performance. The area under the ROC curve was similar to the values reported in other studies for different patient populations, between 70% and 80%, as well as the sensitivity and specificity.

Similarly to other ADR risk stratification studies, patients hospitalized in the intensive care unit and patients under chemotherapy were excluded, because in those patients ADR are naturally expected due to the greater complexity of care and the characteristic toxicity of the regimens, respectively. Contrarily to most other studies, we did not include patients previously hospitalized during the study period, preserving thus the condition of independence of observations. As this tool was designed to be applied at any time during a patient's hospital stay, it is not expected that patients have laboratory results available every day. Our results confirmed our assumption that patients with normal laboratory test results do not have them repeated often, so if a result is unavailable at the time the tool is applied it can be inferred that its true value is lower than its cutoff. In order to be able to include laboratory parameters in the ADR risk tool, we used multiple imputation methods for the estimation of model parameters.

This study has some limitations. The research was conducted in a single hospital, which is a medium-sized tertiary care hospital, and this may limit the generalization of the risk score to patients hospitalized in similar clinical settings. The cases and controls were not similarly distributed among hospital wards, which was our initial intent but could not be applied in practice because of the relatively small number of hospital beds and the large length of stays in this high-complexity tertiary care hospital. Another limitation to the generalization of the results might be that the hospital has a standard formulary and, therefore, some existing medications were never administered to the patients. The study was conducted in a university hospital with a clinical pharmacy team that reviews daily all medication orders and proposes pharmaceutical interventions to the healthcare team, which may have resulted in a lower incidence of adverse events, particularly those related to dose adjustments, treatment duration, potential drug-drug interactions and drug incompatibilities, than what is observed in hospitals without this organization. We were unable to collect laboratory data from many patients because of unwarranted interference in the clinical routine, and some laboratory parameters may be important predictors that are missing in the model. Although we used an objective method to find the cut-off values for dichotomization of the interval-scaled variables, they were based on our patient sample and may not represent the best and most reproducible cut-off values for those variables. We have not performed an external validation of the ADR risk tool in different hospitals and different patient populations.

The main strengths of this study are the prospective design, the large cohort size, the long observation period, the active search for ADR and the internal validation of the score by different methods. This is also the first study to develop an ADR predictive tool for patients hospitalized in a general hospital and demonstrated the feasibility of scoring systems that identify patients at risk of an impending ADR in that population.

The promising results of the internal validation of this ADR risk stratification tool, associated to the ease of use in clinical practice since all items can be collected at the bedside, indicate that further efforts should be undertaken to identify additional risk factors and to perfect ADR risk tools. Although the tool presented here showed interesting performance statistics, we recognize that this is groundbreaking research on this topic that showed that, despite the challenge of predicting an event that occurs at a rate of only about 5% in a heterogeneous patient population, it is possible to identify patients at risk of ADR. These results should stimulate further research in this topic.

## Conclusions

Female, previous history of ADR, heart rate $\geq$ 72 bpm, systolic blood pressure $\geq$148 mmHg, diastolic blood pressure $<$ 79 mmHg, diabetes mellitus, serum urea $\geq$ 67 mg/dL, serum sodium $\geq$ 141 mmol/L, serum potassium $\geq$ 4.9 mmol/L, main diagnosis classified in ICD-10 chapter II—Neoplasms, prescription of $\geq$ 3 ATC class B drugs—blood and blood forming organs, prescription of ATC class R drugs—respiratory system, prescription of intravenous drugs, and prescription of $\geq$ 6 oral drugs, are risk factors of ADR in inpatients of a general hospital who are not in an ICU or undergoing chemotherapy. A risk stratification instrument based on these 14 risk factors showed, in the internal validation, an area under the ROC curve of 0.73, sensitivity of 61% and specificity of 73%.

## Supporting information

**S1 File. Data file.**
(XLSX)

**S1 Table. Supplementary table.**
(DOCX)

## Acknowledgments

The authors acknowledge the essential participation of the students who contributed to the research Kelyanne M Cunha, Milena J Santos, Amanda M Souza, Jézica M Sucar, Jéssica F Ferreira, Maria D Santos, Hárgila H Carvalho, and Karen D Araújo.

## Author Contributions

**Conceptualization:** Sara Iasmin Vieira Cunha Lima, Valdjane Saldanha, Antonio Gouveia Oliveira.

**Data curation:** Rand Randall Martins, Vivian Nogueira Silbiger, Isabelle Cristina Clemente dos Santos, Antonio Gouveia Oliveira.

**Formal analysis:** Antonio Gouveia Oliveira.

**Investigation:** Sara Iasmin Vieira Cunha Lima.

**Methodology:** Rand Randall Martins, Valdjane Saldanha.

**Project administration:** Sara Iasmin Vieira Cunha Lima.

**Resources:** Vivian Nogueira Silbiger, Isabelle Cristina Clemente dos Santos, Ivonete Batista de Araújo.

**Supervision:** Ivonete Batista de Araújo.

**Validation:** Antonio Gouveia Oliveira.

**Visualization:** Sara Iasmin Vieira Cunha Lima.

**Writing – original draft:** Sara Iasmin Vieira Cunha Lima.

**Writing – review & editing:** Vivian Nogueira Silbiger, Ivonete Batista de Araújo, Antonio Gouveia Oliveira.

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
