## [Decision Letter · Decision Letter 0]

9 Mar 2020

PONE-D-19-28832

Development and validation of a clinical instrument for risk assessment of adverse drug reactions in hospitalized patients

PLOS ONE

Dear Ms. Lima,

Thank you for submitting your manuscript to PLOS ONE. After careful consideration, we feel that it has merit but does not fully meet PLOS ONE’s publication criteria as it currently stands. Therefore, we invite you to submit a revised version of the manuscript that addresses the points raised during the review process.

We would appreciate receiving your revised manuscript by Apr 13 2020 11:59PM. To enhance the reproducibility of your results, we recommend that if applicable you deposit your laboratory protocols in protocols.io, where a protocol can be assigned its own identifier (DOI) such that it can be cited independently in the future. For instructions see: http://journals.plos.org/plosone/s/submission-guidelines#loc-laboratory-protocols

We look forward to receiving your revised manuscript.

Kind regards,

Karen Cohen

Academic Editor

PLOS ONE

Journal Requirements:

Please provide an amended Funding Statement that declares *all* the funding or sources of support received during this specific study (whether external or internal to your organization) as detailed online in our guide for authors at http://journals.plos.org/plosone/s/submit-now.  Please state what role the funders took in the study.  If any authors received a salary from any of your funders, please state which authors and which funder. If the funders had no role, please state: "The funders had no role in study design, data collection and analysis, decision to publish, or preparation of the manuscript."

5. Please upload a copy of Figure 1, to which you refer in your text on page 11. If the figure is no longer to be included as part of the submission please remove all reference to it within the text.

6. Please include a caption for figure 1.

Reviewers' comments:

Reviewer's Responses to Questions

**Comments to the Author**

1. Is the manuscript technically sound, and do the data support the conclusions?

Reviewer #1: Yes

Reviewer #2: Partly

Reviewer #3: Yes

2. Has the statistical analysis been performed appropriately and rigorously? 

Reviewer #1: Yes

Reviewer #2: Yes

Reviewer #3: Yes

3. Have the authors made all data underlying the findings in their manuscript fully available?

Reviewer #1: Yes

Reviewer #2: Yes

Reviewer #3: No

4. Is the manuscript presented in an intelligible fashion and written in standard English?

Reviewer #1: Yes

Reviewer #2: No

Reviewer #3: No

5. Review Comments to the Author

Reviewer #1: An observational case to control matched study was conducted to identify predictors of adverse drug reactions and develop a predictive scale of risk. Stepwise conditional logistic regression was used to predict ADR. The training set included ⅔ the sample and the validation set the other ⅓. The validation subset indicated good calibration and discrimination.

Minor revisions:

Line 138: Indicate the statistical testing method which achieves 80% power and if the alpha level was one- or two-sided.

Reviewer #2: Has the statistical analysis been performed appropriately and rigorously? (Answer options: Yes, No, I don't know, N/A)

The study would benefit

The paper would benefit from editing in terms of grammar and language as the authors’ intent is quite unclear in parts.

The identification of ADRS is a clinically difficult and subjective process and it is not clear how the authors dealt with this.

Was the process for validation of the ADRs-was double verification used?

Were patients who presented with an ADR included in the analysis?

How were repeat admissions dealt with for a single patient (in other studies previous ADR has been shown to be a risk factor for future ADRs)

Was length of stay the only matching variable for matching cases to controls?

Given the number of potential variables for consideration for the model the training set appears significantly under powered. More information regarding the power calculation is required as 230 cases appears small given the number of potential variables explored.

The clinical relevance of the algorithm is difficult to ascertain given the large number of variables required.

Reviewer #3: Manuscript Number: PONE-D-19-28832

Full Title: Development and validation of a clinical instrument for risk assessment of adverse drug reactions in hospitalized patients

General comment:

This is a case-control design nested in an open cohort of 8060 patients from various age groups admitted to single tertiary care hospital in Brazil. The author developed a clinical instrument to predict the occurrence of adverse drug reactions (ADR) and performed internal validation by performing data splitting. The performance (prediction accuracy) of the clinical instrument were assessed through calculating the accuracy of score (sensitivity and specificity) and discrimination of the model using AUROC.

Although there have been several risk score have been published earlier, the current study identified several newly identified risk factors which have not been published previously. The author also claimed that their choice of study design contributes to more realistic due to varying nature of clinical condition in an acute patient.

In general, the manuscript is moderately written and have adopted appropriate statistical method to develop and validate a risk prediction tool. However, there are some weakness identified in the method section. One of the concern which have not been highlighted in the limitation is the reliability of the identification and classification of the suspected ADR in the ward. However, the author did acknowledge the generalizability issue which are expected from this clinical instrument since it was only developed based on data from one centre and was not validated in another population or dataset. There are many statements provided in this manuscript without citing any references. The author should review this aspect.

While the author is to be commended for his effort, there are a number of questions remaining to be clarified and are outlined below by section.

Title

The title is suitable and reflect the main content of the manuscript.

Abstract

• Line 27 - It is not clear what the author meant by ‘initial’ validation? Perhaps it will be clearer if the author explicitly mention whether it is internal or external validation.

• Line 27 – The choice of terminology must be standardized throughout the manuscript. The term ‘clinical questionnaire’ were used in the abstract but ‘clinical instrument’ were used in the main text. Moreover, other terminologies were used interchangeably such as ‘predictive scale’ and ‘model of risk prediction’.

• Line 30 – 18-month

Introduction

• Line 66 – It is not clear what does it meant by partial or total validation.

• Line 69 – Perhaps it will be good to highlight in the sentence whether the limiting applicability of the existing model are for patients hospitalised in hospital, ambulatory care setting or general population who are taking medications.

• Line 73 – change ‘initial’ to internal validation.

Method

• Line 79 – It will be good to mention the ‘total number of beds’ rather than vaguely referring as medium-size university hospital.

• Line 81 – General medical ward was not included in the study?

• Line 84 – Patients admitted to oncology wards were included but those on chemotherapy were excluded?

• Line 95 - Potential information bias - not clear how many clinical pharmacists were involved and what are the measures taken to ensure there is reliability in data collected? It will be good if another independent reviewer evaluated each of the identified ADR to increase the reliability of the finding. There are several studies have been published showing the existence of variability between healthcare professionals and also in the method utilised for the identification of ADRs.

• Line 95 – the patients were admitted to clinics or wards?

• Line 101 – Who performed the causality assessment and other classification?

• Line 114 – It will be good if the author provide the rationale for selecting length of hospital stay (LOS) as the matching criteria for control group. The average LOS stay should be reported in the result section.

• Line 123 – What is the accuracy of collecting alcohol intake in grams? If it is estimation or patient reported, it should be mentioned explicitly. Definition of variables selected as risk factors should be clear to ensure high validity in the data collected.

• Line 128 – Provide definition of kidney disease and mild liver impairment.

• Line 136 – Cite reference for the classification of drug interactions.

• Line 140 – How was the data divided randomly? It was mentioned in Line 142 that the training set was ‘planned to have 230 cases’. How was this done randomly? Please provide evidence for this approach.

• Line 144 – No reference were cited for the choice of sample size calculation method utilised in this study.

• Line 150 – Please clarify whether the univariate analysis was performed using the total training and validation set patients. Please provide reference to support the justification for including the entire study population.

• Line 151- The dependent variable should be defined as the occurrence of ADR during hospital stay.

• Line 152 – This should be mentioned in the limitation section.

• Line 173 – No reference provided for the formula used in calculating the estimation of individual probability.

• Line 178 – Provide reference to support the method utilised for the determination of the cut-of value.

• Line 180 – It will be good for readers to know the types of validation conducted in this study.

• Multicollinearity test between the variables included in the final model was not presented.

Results

• Line 188 – How was the cumulative incidence were calculated. Is it at patient level?

• Table 1 – Include the total number of sample included in each control and cases in the first row – frequency value.

• It will be good to perform statistical analysis comparing the case and control so that we can see whether both groups are comparable and there is no selection bias. For example, the number of previous admission is higher in control group as compared to case. Risk of ADR could be higher in patients with multiple history of hospital admission which may contribute to the use of multiple medications. Perhaps, it will be good to discuss this point in discussion and also highlight in the limitation, if necessary.

• Table 2 – It is not clear whether the Haemoglobin, Urea and potassium value was taken on admission, during hospital stay or before discharge.

• Table 2 – The confidence interval is wide for Mental and Behavioural disorders which may provide statistically artefact value in the final model. Suggest to provide supplement table with frequency of cases for each variables included in both case and control.

• Spelling error – Risk C drug interaction

• Line 225 – Provide reference for this statement.

• Line 228 – Score of 4 was selected based on highest sensitivity and specificity? According to this criteria all patients with diabetes with TOD or diagnosis of Mental and Behavioural disorders will be automatically at higher risk. Perhaps, can discuss on this aspect in discussion.

• Table 2 and Table 3: provide information on how the kidney disease and mild liver were defined in this study.

Discussion

• Line 249 – How the author define ‘recent’ model? There are many models have not been included in the current discussion.

• Line 251 – Inaccurate information cited. The inclusion criteria for the development of the BADRI score by Tangiisuran and colleagues was patients more than and equal to 65 years and not 85 years as mentioned. 85 was the median age of patients included in their study.

• Line 252 – Gerontonet Risk Score have been validated in many external cohort. For example, there is an article published validating the risk score in CRIME Cohort. Please conduct comprehensive review of evidences.

• Line 259 – Discuss diastolic blood pressure as significant factor contributing to ADR in your study?

• Line 308 - Based on all the comments highlighted in the comments above, the limitations paragraph may need to be expanded regarding potential threats to internal and external validity.

6. PLOS authors have the option to publish the peer review history of their article (what does this mean?). If published, this will include your full peer review and any attached files.

Reviewer #1: No

Reviewer #2: No

Reviewer #3: Yes: Balamurugan Tangiisuran

---

## [Author Response · Author response to Decision Letter 0]

23 Apr 2020

Response to Reviewers

We are greatly indebted to the Reviewers by their thorough review of the manuscript and the may valuable comments and suggestions. Actually, those comments provoked several reflections that led to an important review of the methodology in order to eliminate as much as possible any subjective elements in the statistical analysis. So we adopted a different method for dichotomizing continuous variables and for setting the cut-off for the risk score, we included laboratory tests that had been collected in at least 85% of patients and used multiple imputation methods to enable their inclusion in the regression models, and clarified many points of the manuscript that were suggested by the Reviewers. This led to the proposal of a different ADR risk tool, but we believe that is more accurate and robust than the previous one. We accepted all the comments and made the appropriate changes in the manuscript, that we detail in this response to the reviewers. In addition, we need assistance in the new analysis and interpretation of the laboratory parameters entered. This implied the inclusion of two co-authors (Silbiger, VN and Santos, ICC). All line and page information refer to the revised manuscript without tracked changes.

Response:

The section “Results” (line 238) was corrected to level 1 heading; the figure 1 citation (line 304 and 317) and figure 2 citation (line 321) was corrected to “Fig 1” and “Fig 2”; and the figures captions (line 312 and 328) was entered as Fig 1. ROC curves of the ADR risk stratification tool in the development sample (dashed line) and in the validation sample (solid line). and Fig 2. ROC curves of the ADR risk stratification tool with unavailable laboratory values scored 0 in the development sample (dashed line) and in the validation sample (solid line).

Response: We revised the text for English language. 

Response: One of the authors (A.G.O.) was responsible for reviewing the text. This author has published over 100 papers and two editions of a book in English language and was a U.S. resident for several years.

Response: We clarify that this study received no funding.

• Please provide an amended Funding Statement that declares *all* the funding or sources of support received during this specific study (whether external or internal to your organization) as detailed online in our guide for authors at http://journals.plos.org/plosone/s/submit-now. 

• Please state what role the funders took in the study. If any authors received a salary from any of your funders, please state which authors and which funder. If the funders had no role, please state: "The funders had no role in study design, data collection and analysis, decision to publish, or preparation of the manuscript."

Response: This study was not funded, It only received financial support for publication. The following text was added to the Acknowledgments section (line 492, page 24):

 “This study was financed in part by the Coordenação de Aperfeiçoamento de Pessoal de Nível Superior - Brasil (CAPES) – Finance Code 001”

Response: As requested, we make the database available as supplementary material.

5. Please upload a copy of Figure 1, to which you refer in your text on page 11. If the figure is no longer to be included as part of the submission please remove all reference to it within the text.

Response: Figures 1 and 2 were uploaded.

6. Please include a caption for figure 1.

Response: The following figure legends were included:

Fig 1. ROC curves of the ADR risk stratification tool in the development sample (dashed line) and in the validation sample (solid line).

Fig 2. ROC curves of the ADR risk stratification tool with unavailable laboratory values scored 0 in the development sample (dashed line) and in the validation sample (solid line).

Reviewers' comments:

Reviewer #1: 

An observational case to control matched study was conducted to identify predictors of adverse drug reactions and develop a predictive scale of risk. Stepwise conditional logistic regression was used to predict ADR. The training set included ⅔ the sample and the validation set the other ⅓. The validation subset indicated good calibration and discrimination.

Minor revisions:

1. Line 138: Indicate the statistical testing method which achieves 80% power and if the alpha level was one- or two-sided.

Response: The text was changed to (line 164, page 7):

“A sample size of 340 cases would afford 80% power, at a significance level of 5%, to identify associations, by conditional logistic regression in a matched case-control study with a 2:1 ratio of controls to cases, with an odds-ratio (OR) of ADR of 1.7 in variables occurring in 5% of the controls to an OR of 1.3 in variables occurring in 50% of the controls.”

Reviewer #2: 

1. The paper would benefit from editing in terms of grammar and language as the authors’ intent is quite unclear in parts.

Response: We revised the text for English language. 

2. The identification of ADRS is a clinically difficult and subjective process and it is not clear how the authors dealt with this. Was the process for validation of the ADRs-was double verification used?

Response: All suspected ADR were analyzed between at least two researchers. In case of doubt, a third researcher was consulted. A clarification has been added to the text in this regard in line 103, page 5.

The previous text was:

“The identification of ADR was performed daily by clinical pharmacists through active search in all the patients hospitalized in the clinics eligible for this study.”

Changed to:

“The identification of suspected ADR was carried out through an active daily search in all patients admitted to the wards eligible for this study. The search was conducted by three clinical pharmacists (S.I.V.C.L., V.S., I.B.A.), with the help of four pharmacy students previously trained, and all suspected ADRs were evaluated among two researchers (S.I.V.C.L., I.B.A.). In case of doubt, a third party (V.S.) was consulted.” 

3. Were patients who presented with an ADR included in the analysis?

Response: The sample included patients who had been in hospital for more than 24 hours, so if the reason for hospitalization was an ADR, it was not included. This explanation has been added to the text in line 87, page 4:

“Patients admitted because of an ADR, hospitalized in the intensive care unit (ICU), transplanted, under chemotherapy and pregnant women were excluded.”

4. How were repeat admissions dealt with for a single patient (in other studies previous ADR has been shown to be a risk factor for future ADRs)

Response: An explanation was added to ensure better understanding (line 88, page 4).

 “Only the first hospitalization since study start was considered, duplicate hospitalization episodes of the same patients being easily detected because each patient receives a lifetime numeric code that identifies the patient across multiple hospital admissions.”

5. Was length of stay the only matching variable for matching cases to controls?

Response: Yes. We added an explanation for this in the Discussion section, line 396, page 20:

“Nested case-control studies are typically based on closed cohorts, with all subjects admitted to the cohort at the same moment in time, while ours was an open cohort with patients entering the cohort throughout the duration of the study. In order to account for this difference, the matched controls were selected at random from all patients in the eligible wards with approximately the same time as the index case since hospital admission.”

6. Given the number of potential variables for consideration for the model the training set appears significantly under powered. More information regarding the power calculation is required as 230 cases appears small given the number of potential variables explored.

Response: As stated in the Statistical Analysis section, the study sample size affords 80% power to identify risk factors with an odds-ratio of 1.7 or more in univariate analysis. The multivariate analysis included 38 variables and was based on 343 cases, providing an approximate ratio of 10 outcomes to each independent variable, which is the recommended ratio for multivariate analysis. Therefore, we believe the study is adequately powered. We would like to point out that the part of the study conducted with the slip-sample method was just to develop the scoring system of the risk stratification tool, and for internal validation, the selection of the relevant risk factors was conducted in the full sample. We emphasized this in the Statistical Analysis section with the following text, line 203, page 8:

“Therefore, for the development of the scoring system, we used only the data of the development sample and a multivariate binary logistic regression model with clustering on cases and robust standard errors, in which the independent variables were the risk factors previously identified in conditional logistic regression, to estimate the partial regression coefficients.”

7. The clinical relevance of the algorithm is difficult to ascertain given the large number of variables required.

Response: Al the variables required for the ADR risk tool are readily collected at bedside or from the patient record, so we believe it is quite practical. The tool has 14 variables, while other reported tools have between 6 and 8 variables of similar nature. However, they were developed to be applied in much more homogeneous patient populations than ours. We pointed out the applicability of the tool in a paragraph in the Discussion section, line 340, page 18:

“The risk tool score can be easily computed at bedside at any time during the hospital stay, to help identify those patients in a general medical or surgical ward who are at risk of experiencing an ADR.”

Reviewer #3: 

General comment:

This is a case-control design nested in an open cohort of 8060 patients from various age groups admitted to single tertiary care hospital in Brazil. The author developed a clinical instrument to predict the occurrence of adverse drug reactions (ADR) and performed internal validation by performing data splitting. The performance (prediction accuracy) of the clinical instrument were assessed through calculating the accuracy of score (sensitivity and specificity) and discrimination of the model using AUROC.

Although there have been several risk score have been published earlier, the current study identified several newly identified risk factors which have not been published previously. The author also claimed that their choice of study design contributes to more realistic due to varying nature of clinical condition in an acute patient.

In general, the manuscript is moderately written and have adopted appropriate statistical method to develop and validate a risk prediction tool. However, there are some weakness identified in the method section. One of the concern which have not been highlighted in the limitation is the reliability of the identification and classification of the suspected ADR in the ward. However, the author did acknowledge the generalizability issue which are expected from this clinical instrument since it was only developed based on data from one centre and was not validated in another population or dataset. There are many statements provided in this manuscript without citing any references. The author should review this aspect.

While the author is to be commended for his effort, there are a number of questions remaining to be clarified and are outlined below by section.

Title

1. The title is suitable and reflect the main content of the manuscript.

Abstract

2. Line 27 - It is not clear what the author meant by ‘initial’ validation? Perhaps it will be clearer if the author explicitly mention whether it is internal or external validation.

Response: We agree and changed the following text (line 28, page 2):

Objective: Development and initial validation of a clinical questionnaire for the assessment of the risk of adverse drug reactions (ADR) among patients hospitalized in a general hospital.

to:

Objective: Development and internal validation of a clinical questionnaire for the assessment of the risk of adverse drug reactions (ADR) among patients hospitalized in a general hospital.

3. Line 27 – The choice of terminology must be standardized throughout the manuscript. The term ‘clinical questionnaire’ were used in the abstract but ‘clinical instrument’ were used in the main text. Moreover, other terminologies were used interchangeably such as ‘predictive scale’ and ‘model of risk prediction’.

Response: We adopted the term clinical tool throughout the manuscript.

4. Line 30 – 18-month

Response: In order to keep the Abstract within the word count limitations we deleted the reference to the duration of the study.

Introduction

5. Line 66 – It is not clear what does it meant by partial or total validation.

The terms “partially” and “totally” were corrected to “internally” and “externally”. (line 66 page 3).

6. Line 69 – Perhaps it will be good to highlight in the sentence whether the limiting applicability of the existing model are for patients hospitalised in hospital, ambulatory care setting or general population who are taking medications.

Response: The text was corrected, to allow a better understanding (line 65, page 3).

The previous text was:

“A few risk predictions models have also been developed and have been validated, partially or totally, but all were intended for populations with specific characteristics, such as the elderly [8-10] or renal patients [6], limiting the applicability of the instrument. The existence of a practical clinical instrument for risk stratification that could identify hospitalized patients at risk for ADR would be of great value to increase patient safety and to guide the care of the medical team, including the clinical pharmacist.”

Was changed to:

“A few risk predictions models have also been developed and have been validated, internally or externally, but all were intended for populations with specific characteristics, such as the elderly [8-10] or renal patients [6], limiting the applicability of the instrument in inpatients in general. The existence of a practical clinical instrument for risk stratification that could identify patients hospitalized in wards of general hospitals at risk for ADR, regardless of age or specific clinical condition, would be of great value to increase patient safety and to guide the care of the medical and the clinical pharmacy team. Thus, the objective of this study was the identification of risk factors for ADR in patients hospitalized in general hospital and the construction and internal validation of a clinical tool predictive of ADR.”

7. Line 73 – change ‘initial’ to internal validation.

Response: The term was changed (line 75).

Method

8. Line 79 – It will be good to mention the ‘total number of beds’ rather than vaguely referring as medium-size university hospital.

Response: The following text (line 78, page 4):

“This was an observational, analytical, case-control study nested in an open cohort consisting of all patients admitted between June 2016 and December 2017 at the Onofre Lopes University Hospital in Natal, Brazil, a tertiary care medium-size university hospital.”

was changed to:

“This was an observational, analytical, case-control study nested in an open cohort consisting of all patients admitted between June 2016 and December 2017 at the Onofre Lopes University Hospital in Natal, Brazil, a tertiary care medium-size university hospital with 247 beds.”

9. Line 81 – General medical ward was not included in the study?

Response: The hospital is organized in specific clinic wards and there is no general medical ward. For better understanding, the text was corrected (line 81, page 4).

Previous text:

“Patients of both sexes, aged 18 years or over, hospitalized in the departments of neurology, mental health, nephrology, urology, cardiology, oncology, gastroenterology, rheumatology and surgery, with a length of stay of more than 24 hours and administered at least one medication during the hospitalization were included in the study.”

Changed to:

“Male and female patients over 18 years-old, with a length of stay of more than 24 hours and administered at least one medication during the hospitalization were included in the study. Only patients hospitalized in the following departments were included (the hospital is organized in clinical specialty wards and there is no general medical ward): neurology, mental health, nephrology, urology, cardiology, oncology (not receiving chemotherapy), gastroenterology, rheumatology and surgery.”

10. Line 84 – Patients admitted to oncology wards were included but those on chemotherapy were excluded?

Response: Oncologic patients hospitalized for purposes other than chemotherapy, such as surgical patients, were included in the sample. The text was added for better understanding (line 86, page 4).

Previous text:

“Patients of both sexes, aged 18 years or over, hospitalized in the departments of neurology, mental health, nephrology, urology, cardiology, oncology, gastroenterology, rheumatology and surgery (once the hospital is organized in specific clinic wards and there is no general medical ward), with a length of stay of more than 24 hours and administered at least one medication during the hospitalization were included in the study.”

Changed to:

“Only patients hospitalized in the following departments were included (the hospital is organized in clinical specialty wards and there is no general medical ward): neurology, mental health, nephrology, urology, cardiology, oncology (not receiving chemotherapy), gastroenterology, rheumatology and surgery.”

11. Line 95 - Potential information bias - not clear how many clinical pharmacists were involved and what are the measures taken to ensure there is reliability in data collected? It will be good if another independent reviewer evaluated each of the identified ADR to increase the reliability of the finding. There are several studies have been published showing the existence of variability between healthcare professionals and also in the method utilised for the identification of ADRs.

Response: All suspected ADR were analyzed between at least two researchers. In case of doubt, a third researcher was consulted. A clarification has been added to the text in this regard in line 103, page 5.

The previous text was:

“The identification of ADR was performed daily by clinical pharmacists through active search in all the patients hospitalized in the clinics eligible for this study.”

Changed to:

“The identification of suspected ADR was carried out on a daily basis through active search in all patients admitted to the wards eligible for this study. The search was conducted by three clinical pharmacists (S.I.V.C.L, V.S, I.B.A.), helped by four previously trained pharmacy students.”

and:

“The adjudication of ADRs was done independently by two clinical pharmacists of the research team (S.I.V.C.L., I.B.A.), and in case of disagreement, a third researcher (V.S.) was consulted.” 

12. Line 95 – the patients were admitted to clinics or wards?

Response: The following text (line 103, page 5):

“The identification of ADR was performed daily by clinical pharmacists through active search in all the patients hospitalized in the clinics eligible for this study.”

was changed to:

“The identification of suspected ADR was carried out on a daily basis through active search in all patients admitted to the wards eligible for this study.”

13. Line 101 – Who performed the causality assessment and other classification?

Response: The following text was added, line 126, page 5.

“One reseacher (S.I.V.C.L.) was designed to perform all causality and severity classification, in order to reduce bias and standardize the classification.”

14. Line 114 – It will be good if the author provide the rationale for selecting length of hospital stay (LOS) as the matching criteria for control group. The average LOS stay should be reported in the result section.

Response: We added the following text explaining the reason why controls were matched by days since hospital admission (line 396, page 20). Please notice that it was not by length of hospital stay, but by number of days from admission to the ADR.

“Nested case-control studies are typically based on closed cohorts, with all subjects admitted to the cohort at the same moment in time, while ours was an open cohort with patients entering the cohort throughout the duration of the study. In order to account for this difference, the matched controls were selected at random from all patients in the eligible wards with approximately the same time as the index case since hospital admission.”

We have no information on the LOS because case patients left the cohort at the time of the ADR. We did, however, report the average number of days since admission in Table 1.

15. Line 123 – What is the accuracy of collecting alcohol intake in grams? If it is estimation or patient reported, it should be mentioned explicitly. Definition of variables selected as risk factors should be clear to ensure high validity in the data collected.

Response: The following text (line 143, page 6):

“The following variables were collected from all cases and controls at the moment the ADR was detected: age, sex, race, daily intake of alcohol in grams, smoking habits, […]”

was changed to:

“The following variables were collected from all cases and controls at the moment the ADR was detected: age, sex, race, self-reported daily intake of alcohol converted to grams of ethanol, self-reported smoking pack-years, […]”

16. Line 128 – Provide definition of kidney disease and mild liver impairment.

Response: We used the same definitions of comorbidities as in the Charlson’s comorbidity index. We made that clear by changing the following text (line 148, page 6):

“[…] comorbidities (myocardial infarction, chronic heart failure, peripheral vascular disease, cardiovascular disease, dementia, COPD, connective tissue disease, peptic ulcer, diabetes without and with target organ damage, kidney disease, tumor, metastases, mild liver disease, liver disease moderate), […]”

to:

“[…] comorbidities composing the Charlson's Comorbidity Index [15] (myocardial infarction, chronic heart failure, peripheral vascular disease, cardiovascular disease, dementia, COPD, connective tissue disease, peptic ulcer, diabetes mellitus (without and with target organ damage), moderate to severe kidney disease, tumor, metastases, liver disease (mild and moderate), […]”

The following reference was added:

15. Charlson ME, Pompei P, Ales KL, MacKenzie CR. A new method of classifying prognostic comorbidity in longitudinal studies: development and validation. J Chronic Dis. 1987;40(5):373-383.

17. Line 136 – Cite reference for the classification of drug interactions.

Response: The following reference was added:

17. Lexicomp Online. Drug Interactions. Hudson, Ohio: Wolters Kluwer Health, Inc. 2019. Available from: https://online.lexi.com. Last accessed April 14, 2020.

18. Line 140 – How was the data divided randomly? It was mentioned in Line 142 that the training set was ‘planned to have 230 cases’. How was this done randomly? Please provide evidence for this approach.

Response: In order to clarify the split sample process, the following text (line 189, page 8):

“For the development of a predictive scale of the risk of ADR, the study population was randomly divided into a training set, consisting of two-thirds of the cases and the respective controls, and a validation set consisting of the remaining one-third of the observations.”

was changed to:

“For the development of a tool for risk prediction of ADR, the 1/3 split-sample method was used, whereby the patient sample was randomly divided into a development sample, consisting of two-thirds of the cases and the respective controls, and a validation sample consisting of the remaining observations. For this purpose, using a computer random number generator each case was assigned a number and the lower 2/3 were selected for the development sample, along with the matching controls.”

19. Line 144 – No reference were cited for the choice of sample size calculation method utilised in this study.

Response: The following reference was added:

18. Dupont WD. Power calculations for matched case–control studies. Biometrics. 1988; 44(4):1157-1168.

20. Line 150 – Please clarify whether the univariate analysis was performed using the total training and validation set patients. Please provide reference to support the justification for including the entire study population.

Response: That was already in the original submission, in the Statistical Analysis section, as follows (line 174, page 7):

“For the identification of patient variables associated with the occurrence of ADR, univariate analysis of each patient variable by conditional logistic regression was performed initially on the entire study population.”

We rephrased the text as:

“For the identification of risk factors for the occurrence of ADR, we initially performed univariate analysis of each predictor variable using conditional logistic regression on the full study population, the dependent variable being the occurrence of an ADR.”

In addition, we added the following explanatory text in the Discussion section (line 409, page 20) :

“We first obtained evidence, from the multivariate conditional logistic regression model, of the set of variables that are important predictors of ADR. For the identification of the risk factors, we used the full patient sample as has been recommended by several authors, who have criticized the split-sample method for risk factor identification as a waste of data [27,28].”

The following references supporting the inclusion of the entire study population were added:

27. Altman DG, Vergouwe Y, Royston P, Moons KG. Prognosis and prognostic research: validating a prognostic model. BMJ. 2009;338:b605.

28. Moons KGM, Kengne AP, Woodward M, Royston P, Vergouwe Y, Altman DG et al. Risk prediction models: I. Development, internal validation, and assessing the incremental value of a new (bio)marker. Heart. 2012; 98(9):683-690.

21. Line 151- The dependent variable should be defined as the occurrence of ADR during hospital stay.

Response: The following text (line 174, page 7):

“In this model, the dependent variable was the occurrence of ADR.”

was changed to:

“For the identification of risk factors for the occurrence of ADR, we initially performed univariate analysis of each predictor variable using conditional logistic regression on the full study population, the dependent variable being the occurrence of an ADR during hospital stay.”

22. Line 152 – This should be mentioned in the limitation section.

Response: We added the following text in the Discussion section (line 461, page 22):

“Although we used an objective method to find the cut-off values for dichotomization of the interval-scaled variables, they were based on our patient sample and may not represent the best and most reproducible cut-off values for those variables.”

23. Line 173 – No reference provided for the formula used in calculating the estimation of individual probability.

Response: We added the following text in the Statistical Analysis section (line 213, page 9):

“The probability of ADR in each patient is computed as 1/(1+exponential of –Ui), with Ui being the logit of ADR in the ith patient predicted by the regression equation Ui = b0 xib, where x is the set of predictors, each multiplied by the corresponding partial regression coefficient b, and b0 is the regression constant corrected as shown above [20].”

The following reference was added:

20. Hosmer DW, Lemeshow S, Sturdivant RX. Applied logistic regression. (3rd ed.) New York: Wiley. 2013

23. Line 178 – Provide reference to support the method utilised for the determination of the cut-of value.

Response: We changed the method in order to adopt a more usual method of finding cut-offs in ROC curves. The following text (line 169, page 7):

“In order to avoid the problem of the non-normal distribution of many continuous variables, all continuous variables were dichotomized by selecting as cutoff the value corresponding to the maximum sensitivity and specificity determined by ROC curves constructed for each variable”

was changed to:

“In order to avoid the issue of the non-normal distribution of most interval-scaled predictor variables, all such variables were dichotomized by selecting as cutoff the variable value corresponding to the highest value of the Youden’s index computed for each value of the predictor variable [19]. Youden’s index is equal to sensitivity + specificity – 1.”

The following reference was added:

19. Youden WJ. Index for rating diagnostic tests. Cancer. 1950;3(1):32-35.

24. Line 180 – It will be good for readers to know the types of validation conducted in this study.

Response: The following paragraph (line 229, page 9):

“The instrument thus developed was tested in the validation set to estimate its sensitivity and specificity with 95% confidence intervals.”

was changed to:

“For assessment of the internal validation of the ADR risk stratification tool thus developed, the sensitivity, specificity, and AUC of the scoring system were obtained in the validation sample.”

25. Multicollinearity test between the variables included in the final model was not presented.

Response: The following text was included in the Statistical Analysis section (line 181, page 8):

“Variables that had a prevalence among the cases less than 5% and those that exhibited collinearity with other variables, defined as a tetrachoric correlation coefficient ≥ 0.70, were excluded.”

In the Results section, we added the following text (line 276, page 12):

“There was no evidence of collinearity among those variables, the median value of the tetrachoric coefficient, a measure of correlation between binary variables, was 0.10 and the maximum value was 0.56.”

Results

26. Line 188 – How was the cumulative incidence were calculated. Is it at patient level?

Response: As we have clarified before that the study was based in distinct patients, it should be clear now that the incidence is at the patients level, not at hospitalization episode level.

27. Table 1 – Include the total number of sample included in each control and cases in the first row – frequency value.

Response: Modified as suggested in table 01.

28. It will be good to perform statistical analysis comparing the case and control so that we can see whether both groups are comparable and there is no selection bias. For example, the number of previous admission is higher in control group as compared to case. Risk of ADR could be higher in patients with multiple history of hospital admission which may contribute to the use of multiple medications. Perhaps, it will be good to discuss this point in discussion and also highlight in the limitation, if necessary.

Response: It must be kept in mind that it is not expected that the groups are similar in the distribution of potential risk factors, it is exactly the opposed. They should not be too different in variables related to the conduct of the study, which in this case would be the time since admission, the death rate and the distribution by wards. Therefore, we added the following text in the Results section related only to those variables (line 251, page 10):

“There were no statistically significant differences between groups in the time since admission (p=0.54) and in-hospital death (p=0.39). However, the distribution by hospital wards was statistically different (p=0.001).”

We also added the following text in the limitations paragraph of the Discussion section (line 448, page 22):

“The cases and controls were not similarly distributed among hospital wards, which was our initial intent but could not be applied in practice because of the relatively small number of hospital beds and the large length of stays in this high-complexity tertiary care hospital.”

29. Table 2 – It is not clear whether the Haemoglobin, Urea and potassium value was taken on admission, during hospital stay or before discharge.

Response: The following text was added as a footnote of Table 2:

“All variables were collected at the time of the ADR in the cases and of randomization in the controls.”

30. Table 2 – The confidence interval is wide for Mental and Behavioural disorders which may provide statistically artefact value in the final model. Suggest to provide supplement table with frequency of cases for each variables included in both case and control.

Response: We agree, that was because the prevalence of that diagnosis was less than 3% among the cases. We recognize the problem of including infrequently observed variables in the model, so we excluded all variables with a prevalence below 5% among the cases. We included a note in the Statistical Analysis section by adding the following text (line 181, page 8):

“Variables that had a prevalence among the cases less than 5% and those that exhibited collinearity with other variables, defined as a tetrachoric correlation coefficient ≥ 0.70, were excluded.”

In addition, we added the following text in the Results section (line 260, page 11):

“For multivariate analysis, 12 variables were excluded, because of collinearity (height, creatinine, type C drug-drug interactions, prescription of ATC class A drugs, and number of medications), prevalence among cases less than 5% (serum bicarbonate, prescription of ATC class H drugs, and main diagnosis of ICD-10 chapter V), missing data (albumin, gamma GT and alkaline phosphatase) and Charlson’s comorbidity index to avoid having a score within a score.”

31. Spelling error – Risk C drug interaction

Response: The error has been corrected in Table 02.

32. Line 225 – Provide reference for this statement. (The brier score)

Response: The following reference was added:

21. Brier GW. Verification of forecasts expressed in terms of probability. Mon Weather Rev. 1950; 78(1): 1–3.

33. Line 228 – Score of 4 was selected based on highest sensitivity and specificity? According to this criteria all patients with diabetes with TOD or diagnosis of Mental and Behavioural disorders will be automatically at higher risk. Perhaps, can discuss on this aspect in discussion.

Response: That was correct and was because the incidence of ADR in patients with those variables is extremely high. However, we decided to exclude diagnosis of mental disorder from the model because it was present in a very small number of patients, and we decided to eliminate the distinction of diabetes in with and without organ damage because in retrospect it seemed difficult to ensure a complete assessment of end-organ damage in many patients. Therefore we considered only a diagnosis of diabetes, which is less prone to errors. In this way, that issue is no longer relevant.

34. Table 2 and Table 3: provide information on how the kidney disease and mild liver were defined in this study.

Response: As mentioned in a previous response, we referenced the definitions of kidney and liver disease to the ones set forth in Charlson’s paper in this way (line 148, page 6):

“…comorbidities composing the Charlson's Comorbidity Index [15] (myocardial infarction, chronic heart failure, peripheral vascular disease, cardiovascular disease, dementia, COPD, connective tissue disease, peptic ulcer, diabetes mellitus (without and with target organ damage), moderate to severe kidney disease, tumor, metastases, liver disease (mild and moderate)...”

Discussion

35. Line 249 – How the author define ‘recent’ model? There are many models have not been included in the current discussion.

Response: The way the sentence was arranged, incorrectly limited the number of existing models. Therefore, the sentence was corrected from (line 344, page 18):

“In the literature we have found only five recent models of risk prediction of ADR”

To:

“We have found four models published the last 10 years on risk prediction of ADR.” 

36. Line 251 – Inaccurate information cited. The inclusion criteria for the development of the BADRI score by Tangiisuran and colleagues was patients more than and equal to 65 years and not 85 years as mentioned. 85 was the median age of patients included in their study.

Response: We recognize our error and apologize for that. We changed the following text (line 345, page 18):

“In all of them the predictive models were developed for specific populations, namely elderly patients over 65 years-old [8,9], over 85 years-old [10], patients with chronic kidney disease [6] and surgical patients [16].”

to:

“In all of them the predictive models were developed for specific populations, namely elderly patients [8,10], patients with chronic kidney disease [6], and surgical patients [20].”

37. Line 252 – Gerontonet Risk Score have been validated in many external cohort. For example, there is an article published validating the risk score in CRIME Cohort. Please conduct comprehensive review of evidences.

Response:

We apologize for the mistake. The information has been corrected from (line 347, page 18):

“Those studies presented only an initial validation based only in the report of the area under the ROC curve or presented low specificity.”

To:

“Two risk prediction tools, the Gerontonet Risk Score [8] and the BADRI Model [10], were validated in an external cohort [9]. The other two presented results only of the internal validation based on bootstrap estimates on the entire patient sample.”

38. Line 259 – Discuss diastolic blood pressure as significant factor contributing to ADR in your study?

Response: We prefer not to be too speculative, as ADR have a complex etiology and no single explanation is likely to be correct. We preferred to discuss our findings in the light of results reported by other studies. Therefore, we changed to following paragraph (line 356, page 18):

 “Vascular disease has been described as a risk factor for ADR [6], but diastolic blood pressure was not found in the literature review as an independent risk factor.”

to: 

“Vascular disease was described as a risk factor for ADR [6] but low diastolic blood pressure has not previously been identified as an independent risk factor, although risk factors related to hypotension have been reported, namely administration of anti-hypertensive and diuretics [23].”

39. Line 308 - Based on all the comments highlighted in the comments above, the limitations paragraph may need to be expanded regarding potential threats to internal and external validity.

Response: We changed the limitations paragraph from (line 446, page 22):

“This study has some limitations. The research was conducted in a single hospital, which is a medium-sized tertiary care hospital, and this may limit the generalization of the risk score to patients hospitalized in similar clinical settings. Another limitation to the generalization of the results might be that the hospital has a standard formulary and, therefore, some existing medications were never administered in the patients. Also, the study was conducted in a university hospital with a clinical pharmacy team that reviews daily all medical orders and that proposes pharmaceutical interventions to the healthcare team, and this may have resulted in a lower incidence of adverse events than observed in hospitals without this organization. The validation of the instrument was based on bootstrapping resampling and in the application to a different patient set, and requires more extensive validation in different populations and other clinical settings. The main strengths of this study are the prospective design, the large cohort size, the long observation period, the active search for ADR and the validation of the score by different methods.”

to:

“This study has some limitations. The research was conducted in a single hospital, which is a medium-sized tertiary care hospital, and this may limit the generalization of the risk score to patients hospitalized in similar clinical settings. The cases and controls were not similarly distributed among hospital wards, which was our initial intent but could not be applied in practice because of the relatively small number of hospital beds and the large length of stays in this high-complexity tertiary care hospital. Another limitation to the generalization of the results might be that the hospital has a standard formulary and, therefore, some existing medications were never administered to the patients. The study was conducted in a university hospital with a clinical pharmacy team that reviews daily all medication orders and proposes pharmaceutical interventions to the healthcare team, which may have resulted in a lower incidence of adverse events, particularly those related to dose adjustments, treatment duration, potential drug-drug interactions and drug incompatibilities, than what is observed in hospitals without this organization. We were unable to collect laboratory data from many patients because of unwarranted interference in the clinical routine, and some laboratory parameters may be important predictors that are missing in the model. Although we used an objective method to find the cut-off values for dichotomization of the interval-scaled variables, they were based on our patient sample and may not represent the best and most reproducible cut-off values for those variables. We have not performed an external validation of the ADR risk tool in different hospitals and different patient populations.”

---

## [Decision Letter · Decision Letter 1]

15 Oct 2020

PONE-D-19-28832R1

Development and validation of a clinical instrument for risk assessment of adverse drug reactions in hospitalized patients

PLOS ONE

Dear Dr. Lima,

Thank you for submitting your manuscript to PLOS ONE. After careful consideration, we feel that it has merit but does not fully meet PLOS ONE’s publication criteria as it currently stands. Therefore, we invite you to submit a revised version of the manuscript that addresses the points raised during the review process.

We look forward to receiving your revised manuscript.

Kind regards,

Karen Cohen

Academic Editor

PLOS ONE

Additional Editor Comments (if provided):

Please can the authors address the following issues:

1. Title: The current title is unclear. The tool is not for "risk assessment of adverse reactions" but rather a tool to predict risk of an adverse reaction. Suggest revise to make this clearer, for example: Development and validation of a clinical instrument to predict risk of an adverse drug reaction in hospitalised patients.

2. There is no information included in the paper as to what ADRs were identified, which organ system was implicated, and what the implicated medicines were. Please can the authors include some description of the ADRs that were identified in the cases, and the medicines implicated in these adverse reactions. A supplementary table detailing the ADRs identified in the case, and implicated drugs. would be very useful.

3. Some comment on the prevalent ADRs in the cases should be included. This is important for assessment of generalizability and clinical utility of this instrument.

4. In response to a reviewer's comment on table 2, the following text was added to table 2 in this revision: “All variables were collected at the time of the ADR in the cases and of randomization in the controls.”

The controls were not randomized, so it is unclear what this refers to. Please correct this footnote.

Reviewers' comments:

Reviewer's Responses to Questions

**Comments to the Author**

Reviewer #1: All comments have been addressed

Reviewer #3: All comments have been addressed

2. Is the manuscript technically sound, and do the data support the conclusions?

Reviewer #1: (No Response)

Reviewer #3: Yes

3. Has the statistical analysis been performed appropriately and rigorously? 

Reviewer #1: (No Response)

Reviewer #3: Yes

4. Have the authors made all data underlying the findings in their manuscript fully available?

Reviewer #1: (No Response)

Reviewer #3: Yes

5. Is the manuscript presented in an intelligible fashion and written in standard English?

Reviewer #1: (No Response)

Reviewer #3: Yes

6. Review Comments to the Author

Reviewer #1: (No Response)

Reviewer #3: The authors have adequately addressed all the comments raised in a previous round of review with satisfactory explanation and response. I recommend that this manuscript is now acceptable for publication.

7. PLOS authors have the option to publish the peer review history of their article (what does this mean?). If published, this will include your full peer review and any attached files.

Reviewer #1: No

Reviewer #3: **Yes: **Balamurugan Tangiisuran

---

## [Author Response · Author response to Decision Letter 1]

3 Nov 2020

We are greatly indebted to the Editor by his careful review of the manuscript and the valuable suggestions. We accepted all the comments and made the appropriate changes in the manuscript, that we detail in this rebuttal letter to the academic editor.

1. Title: The current title is unclear. The tool is not for "risk assessment of adverse reactions" but rather a tool to predict risk of an adverse reaction. Suggest revise to make this clearer, for example: Development and validation of a clinical instrument to predict risk of an adverse drug reaction in hospitalised patients.

Response: We changed the title, as requested, from:

“Development and validation of a clinical instrument for risk assessment of an adverse drug reactions in hospitalized patients”

To:

“Development and validation of a clinical instrument to predict risk of an adverse drug reactions in hospitalized patients”

2. There is no information included in the paper as to what ADRs were identified, which organ system was implicated, and what the implicated medicines were. Please can the authors include some description of the ADRs that were identified in the cases, and the medicines implicated in these adverse reactions. A supplementary table detailing the ADRs identified in the case, and implicated drugs. would be very useful.

Response: We have added in the text (line 252, page 11) information about the most frequent adverse reactions and the medications involved. We prepared a table with all the reactions collected in the patient cases. The table was attached as a supplementary table and included in Supporting Information section (line 603, page 28).

“Hypoglycemia related to insulin administration was the most common ADR (87, 25.4%), followed by hypotension (68, 19.8%) due to the use of different pharmacological classes of antihypertensive, highlighting the angiotensin-converting enzyme inhibitors (26, 7.6%) and loop diuretics (14, 4.1%). The occurrence of nausea and vomiting was observed in 13.1% (45) of the cases, related to several pharmacological classes such as opioid analgesics (18, 5.2%), general anesthetics (4, 1.2%) and laxatives (4, 1.2%). The other ADRs and medications involved can be seen in table 02.”

Supplementary Table – Profile of adverse reactions identified and pharmacological classes involved.

3. Some comment on the prevalent ADRs in the cases should be included. This is important for assessment of generalizability and clinical utility of this instrument.

Response: We included a paragraph in the discussion session (line 346, page 20) on the findings of the most frequent reactions and medications involved.

“It is observed that almost all RAM are typical of the main drugs used, such as hypoglycemia and hypotension associated with hypoglycemic and antihypertensive drugs, respectively. Insulin is commonly related to adverse reactions like this in the hospital setting [22]. The drug-dependent dose reactions may occur due to limitations, such as the controlled diet with limited carbohydrates in the hospital environment and bed restriction [23], resulting in the particularity of the individual dose, subject to the need for constant dose adjustments [24, 25]. Hypotension also seems to be related to the patient's fragility, length of hospital stay and, mainly, age and concomitant use of more than one antihypertensive [26, 27].”

4. In response to a reviewer's comment on table 2, the following text was added to table 2 in this revision: “All variables were collected at the time of the ADR in the cases and of randomization in the controls.”

The controls were not randomized, so it is unclear what this refers to. Please correct this footnote.

Response: The controls were randomized, as explained in Methods section, line 126, page 6: “All patients with suspected ADR were included in the study as cases at the time of the event. The same day, two controls were randomly selected among all patients hospitalized in the eligible wards hospitalized within 5 days of the admission date of the case.”.

---

## [Decision Letter · Decision Letter 2]

30 Nov 2020

Development and validation of a clinical instrument to predict risk of an adverse drug reactions in hospitalized patients

PONE-D-19-28832R2

Dear Dr. Lima,

We’re pleased to inform you that your manuscript has been judged scientifically suitable for publication and will be formally accepted for publication once it meets all outstanding technical requirements.

Kind regards,

Antonio Palazón-Bru, PhD

Academic Editor

PLOS ONE

Additional Editor Comments (optional):

Reviewers' comments:

Reviewer's Responses to Questions

**Comments to the Author**

1. If the authors have adequately addressed your comments raised in a previous round of review and you feel that this manuscript is now acceptable for publication, you may indicate that here to bypass the “Comments to the Author” section, enter your conflict of interest statement in the “Confidential to Editor” section, and submit your "Accept" recommendation.

Reviewer #1: All comments have been addressed

2. Is the manuscript technically sound, and do the data support the conclusions?

Reviewer #1: (No Response)

3. Has the statistical analysis been performed appropriately and rigorously? 

Reviewer #1: (No Response)

4. Have the authors made all data underlying the findings in their manuscript fully available?

Reviewer #1: (No Response)

5. Is the manuscript presented in an intelligible fashion and written in standard English?

Reviewer #1: (No Response)

6. Review Comments to the Author

Reviewer #1: (No Response)

7. PLOS authors have the option to publish the peer review history of their article (what does this mean?). If published, this will include your full peer review and any attached files.

Reviewer #1: No

---

## [Editor Report · Acceptance letter]

3 Dec 2020

PONE-D-19-28832R2 

Development and validation of a clinical instrument to predict risk of an adverse drug reactions in hospitalized patients 

Dear Dr. Lima:

I'm pleased to inform you that your manuscript has been deemed suitable for publication in PLOS ONE. Congratulations! Your manuscript is now with our production department. 

Kind regards, 

on behalf of

Dr. Antonio Palazón-Bru 

Academic Editor

PLOS ONE